# Antiplatelet Therapy Mitigates Brain Metastasis Risk in Non-Small Cell Lung Cancer: Insights from a Comprehensive Retrospective Study

**DOI:** 10.3390/cancers17132059

**Published:** 2025-06-20

**Authors:** Carla Martín-Abreu, María García-Gil, Margarita Méndez-Monge, Helga Fariña-Jerónimo, Julio Plata-Bello

**Affiliations:** 1Department of Medical Oncology, Hospital Universitario de Canarias, 38320 La Laguna, Spain; 2Department of Basic Medical Sciences, Faculty of Medicine, University of La Laguna, 38320 La Laguna, Spain; 3Department of Neurosurgery, Hospital Universitario de Canarias, 38320 La Laguna, Spain

**Keywords:** non-small cell lung cancer (NSCLC), brain metastases, antiplatelet therapy, platelet–tumor interaction, real-world evidence

## Abstract

People with lung cancer often face a serious complication: the spread of cancer to the brain. This condition greatly worsens quality of life and survival. In this study, we looked at data from 650 people with lung cancer to see if taking medications that reduce blood clotting—commonly used to prevent heart attacks or strokes—might also lower the risk of brain cancer spread. These medications, mainly aspirin, were more often used by older patients with other health issues. Still, we found that those who took them had a lower chance of developing brain cancer spread during their illness. This effect was even stronger in people with more advanced lung cancer. Those on these medications also lived longer without their cancer getting worse. Notably, none of the people who started these medications shortly after their cancer diagnosis developed brain cancer spread. Our results suggest that these common medications may help slow or prevent the spread of lung cancer to the brain. If confirmed by future studies, this could lead to new ways to improve outcomes for people with lung cancer using safe, widely available treatments.

## 1. Introduction

Non-small cell lung cancer (NSCLC) ranks among the most prevalent malignancies worldwide. According to the World Health Organization (WHO) and the International Agency for Research on Cancer (IARC), Asia reports the highest incidence of NSCLC (~1,300,000 new cases annually), followed by Europe (480,000) and North America (250,000) [1,2]. Despite advances in therapy, NSCLC remains the leading cause of cancer-related mortality, with a 5-year survival rate of approximately 18% [2,3].

Brain metastases represent a critical prognostic factor in NSCLC, occurring in 30% to 64% of patients during the disease course [4,5]. Their incidence increases with disease stage, from 3% in early-stage NSCLC (I-II) to 21% in stage III and 26% in stage IV [6,7]. These metastases significantly worsen prognosis and cause debilitating neurological symptoms, including motor deficits, seizures, cognitive impairment, and intracranial hypertension, all of which severely impact quality of life [8,9,10].

Several factors have been linked to an increased risk of brain metastases in NSCLC, particularly adenocarcinoma histology, advanced tumor stages, and nodal involvement, as well as EGFR or ALK mutations (among others) and younger age [11,12,13,14,15]. Additionally, KRAS mutations, high carcinoembryonic antigen (CEA) levels, and neuron-specific enolase expression have been identified as potential risk factors for brain metastases in NSCLC [16]. Recent studies have supported the role of age, mutational status, and CEA levels as independent predictors of brain metastases and have enabled the identification of high-risk patient groups through multivariable modeling [17]. In EGFR-mutated NSCLC, further evidence suggests that stable extracranial disease and treatment with erlotinib are associated with improved outcomes following brain metastasis diagnosis [18]. Conversely, thrombocytosis has also been associated with worse prognosis in lung cancer, implying a possible connection between platelet levels and disease progression [19,20,21].

A multidisciplinary approach that combines surgery, radiosurgery, targeted therapies, and immunotherapy is recommended to improve outcomes [22,23,24]. However, treatment remains suboptimal due to the challenges posed by the blood–brain barrier and the aggressive nature of metastatic disease [25]. Given the role of platelets in cancer progression and metastasis, antiplatelet therapy has gained attention as a potential modulator of tumor dissemination, warranting further investigation in NSCLC brain metastases.

While primarily involved in hemostasis, thrombosis, and tissue damage response, accumulating evidence links platelets to tumor development and metastasis [26,27,28]. A key mechanism is tumor cell-induced platelet aggregation (TCIPA), where tumor cells trigger platelet activation, thereby promoting immune evasion and metastatic dissemination [29]. By forming a platelet–tumor complex, platelets protect circulating tumor cells (CTCs) from natural killer (NK) cell-mediated destruction, aided by TGF-β and other immunosuppressive factors [30,31,32,33,34]. Beyond immune evasion, platelets facilitate metastasis by enhancing vascular adhesion, extravasation, and metastatic niche formation [32]. Through pro-coagulant factors and P-selectin, platelets promote tumor-endothelial adhesion, aiding both intravasation and extravasation [29,35]. These interactions help establish to the pre-metastatic niche, preparing distant organs for tumor colonization [33].

Additionally, platelets promote tumor growth and angiogenesis by releasing VEGF, PDGF, and EGF, supporting neovascularization and tumor microenvironment remodeling [32,35,36]. Platelets also protect tumor cells from chemotherapy-induced apoptosis, promoting resistance and disease progression [30,37].

Given their pro-metastatic role, targeting platelet function has emerged as a potential therapeutic approach to limit metastasis and improve outcomes. Preclinical evidence supports this concept: Amirkhosravi et al. demonstrated that XV454, a glycoprotein IIb/IIIa antagonist, reduced platelet–tumor binding and pulmonary metastases in mice [38]. By downregulating TCIPA, XV454 enhances immune cell adhesion to the endothelium, facilitating immune infiltration and response [39]. Antiplatelet agents may disrupt platelet-mediated mechanisms by impairing tumor shielding, vascular adhesion, and metastatic niche formation, potentially enhancing the efficacy of conventional cancer therapies [29]. By blocking platelet aggregation and adhesion, they may enhance conventional cancer therapies and serve as metastatic prevention strategies [40].

Several antiplatelet strategies have shown potential antitumor effects. Platelet receptor inhibitors (e.g., P2Y12 and glycoprotein IIb/IIIa antagonists) impair TCIPA, promoting immune-mediated clearance of tumor cells [41]. Meanwhile, COX-1/COX-2 inhibitors (e.g., aspirin) modulate the inflammatory and thrombotic microenvironment, further disrupting platelet–tumor interactions [42,43]. However, their effects on other platelet-mediated mechanisms that facilitate metastasis, such as vascular adhesion, pre-metastatic niche formation, and tumor-supportive signaling, remain incompletely understood and warrant further investigation. The widespread availability and established safety of antiplatelet drugs make them attractive candidates for oncology repurposing. Long-term aspirin use has been associated with reduced metastasis risk in various cancers [44,45,46]. Integrating antiplatelet therapy into multimodal cancer treatments could enhance conventional therapies and offer prophylaxis against metastases [29,47].

Despite growing evidence linking platelets to tumor progression and metastasis, the potential role of antiplatelet therapy in modulating metastatic risk in NSCLC remains unclear. While preclinical studies suggest that inhibiting platelet function can reduce tumor dissemination, clinical data in NSCLC patients are scarce and largely derived from retrospective analyses. Although some epidemiological studies suggest a lower incidence of metastases in patients receiving long-term antiplatelet therapy [44,45,46], evidence remains heterogeneous and inconclusive, particularly regarding brain metastases. Furthermore, the impact of antiplatelet therapy in real-world NSCLC populations, including its potential association with metastatic burden at diagnosis, has not been systematically explored.

Given these uncertainties, this study aims to evaluate whether the exposure to antiplatelet therapy is associated with a reduced risk of NSCLC brain metastases development. To address this, we conducted a retrospective observational study analyzing clinical data from a cohort of NSCLC patients. This research seeks to provide new insights into the potential protective role of antiplatelet agents in metastatic presentation, contributing to a better understanding of their impact in routine clinical practice.

## 2. Methods

### 2.1. Study Type

Retrospective observational study of a cohort of patients with NSCLC diagnosed and treated at a single center.

### 2.2. Ethics

The study obtained approval from the Research Ethics Committee of our center, adhering to all principles of the Helsinki Declaration. Informed consent was not obtained from patients, as the research is based on the analysis of existing data and does not involve additional interventions or follow-ups. This decision aligns with the non-interventional nature of our study design.

### 2.3. Patients

A total of 722 cases of lung neoplasia diagnosed consecutively at our center from January 2015 to December 2018 (minimum 6 years of follow-up) were evaluated. Seventy-two cases of small cell lung carcinoma were excluded, resulting in a final sample of 650 patients. Patient characteristics are recorded in Table 1.

### 2.4. Study Variables

Epidemiological, clinical, and molecular variables were recorded (Table 1). Given the study’s aim to determine the influence of antiplatelet use on the development of brain metastases, these variables were appropriately documented. Additionally, the presence of brain metastases at the time of NSCLC diagnosis was determined, and the timing of antiplatelet therapy was identified, distinguishing between exposure before diagnosis (more than 30 days before diagnosis) or after diagnosis (up to 30 days before diagnosis). If antiplatelet initiation occurred after the appearance of metastatic lesions, the patient was classified as not exposed to antiplatelets (this situation occurred in only one case). The type of antiplatelet therapy was also recorded. In this regard, 75.0% of cases with antiplatelet use were exposed to acetylsalicylic acid, followed by clopidogrel (5.9%), triflusal (2.5%), and cilostazol (0.5%), all in monotherapy. Combinations of the above were present in 16.2% of patients. Due to the heterogeneity in the distribution of this last variable, it was decided not to include it in subsequent analyses.

Brain metastases were diagnosed based on magnetic resonance imaging (MRI), which was performed in patients presenting with neurological symptoms suggestive of central nervous system involvement. Once brain metastases were identified, follow-up brain MRIs were conducted periodically as part of the monitoring strategy. In some patients, brain MRI was included in the initial staging work-up at diagnosis; in those cases, follow-up MRIs were also performed at regular intervals, regardless of the presence of neurological symptoms. This approach was consistent with local clinical practice guidelines during the study period.

Regarding molecular variables, the sample showed heterogeneity in determination (EGFR mutation in 398 patients; ALK mutation in 364 patients; ROS1 expression in 168 patients; PDL1 expression in 121 patients; and BRAF mutation in 51 patients). This variation was because, at the time of diagnosis, such determination was not mandatory or part of the standard clinical evaluation protocol.

Among the prognostic variables used in the study are progression-free survival (PFS), defined as the period during which no evidence of disease progression is observed, and overall survival (OS), defined as the period until the patient dies or the follow-up ends.

### 2.5. Outcome

The primary variable in our study is the percentage of NSCLC patients who develop brain metastases at any point in the disease. As previously mentioned, for specific analyses, a distinction was made between patients with metastasis at the time of diagnosis and those with metastasis during follow-up.

### 2.6. Statistics

Comparative analyses were performed between groups of patients exposed or not exposed to antiplatelet therapy using non-parametric statistical tests (chi-square/Fisher’s exact test for categorical variables; Mann–Whitney U test for continuous variables). Additionally, the log-rank test and Kaplan–Meier curves were used to determine and compare PFS and OS, as well as the mean times to the development of brain metastases in the study groups. The described analysis was complemented with a subgroup analysis (by tumor stage), conducting univariate COX regression for each stage to analyze the role of antiplatelet use as a prognostic factor. Finally, binary logistic regression analysis was performed to identify risk factors associated with the presence of brain metastases at the time of NSCLC diagnosis. In this context, the binary outcome variable was defined as the presence or absence of brain metastases at lung cancer diagnosis. Initially, univariate analyses were performed to identify variables with statistically significant associations. These variables were then included in the multivariate analysis to adjust for potential confounders and assess independent associations. A significance level was set at *p* < 0.05 for all analyses.

## 3. Results

Patients using antiplatelet therapy have more comorbidities but a lower risk of developing brain metastases.

As expected, the antiplatelet-exposed group showed a higher prevalence of cardiovascular (hypertension, heart failure, renal failure) and metabolic (diabetes, dyslipidemia) comorbidities (chi-square, *p* < 0.05). Additionally, the mean age of this patient group was higher than that of those not exposed to antiplatelets (71.66 vs. 65.19; *p* < 0.001) (Table 2). A different distribution in the initial tumor stage was also observed between both groups, with stages 0 and I being more frequent in the group of patients using antiplatelets (32.8% vs. 18.1%) and stage IV more common in patients not exposed to antiplatelets (51.2% vs. 37.3%) (Table 2).

Regarding brain metastasis development, 89 patients (20.0%) in the non-antiplatelet group developed this complication, compared to 14 patients (6.9%) among those receiving antiplatelet therapy (chi-square; *p* < 0.001) (Table 2). Moreover, the mean time to the development of brain metastases was longer in the group of patients exposed to antiplatelet therapy (62.6 vs. 77.5 months; log-rank, *p* < 0.001) (Table 2; Figure 1a).

Finally, patients using antiplatelet therapy showed a longer PFS (49.9 vs. 38.7 months; log-rank; *p* = 0.002) (Figure 1b, Table 2). Although a higher OS was observed in this group (10.6 vs. 8.2 months), this difference did not reach statistical significance (log-rank; *p* = 0.089) (Figure 1c; Table 2).

### 3.1. Patients Most Benefiting from the Effects of Antiplatelet Therapy Are Those with Advanced Stages at the Time of Diagnosis

Given the differing distribution of tumor stages between patients in the groups with and without exposure to antiplatelet therapy, a subgroup analysis by stages was conducted to evaluate the effect of antiplatelet exposure on the risk of developing brain metastases at any point in the disease. Among patients diagnosed with stages 0 and I (147 patients, 66 exposed to antiplatelet therapy), two cases of brain metastases were recorded, one in each group (chi-square, *p* = 0.891; HR = 1.25, 95% C.I. [0.08–20.09]; *p* = 0.872) (Figure 2a). Among patients with stage II (43 patients, 12 exposed to antiplatelet therapy), four cases of brain metastases were identified. Only one of the cases was exposed to antiplatelet therapy (chi-square; *p* = 0.892; HR = 0.84, 95% C.I. [0.09–9.12]; *p* = 0.882) (Figure 2a). In patients with stage III (154 patients, 49 exposed to antiplatelet therapy), 15 cases of brain metastases were diagnosed, with three belonging to the antiplatelet-exposed group (chi-square, *p* = 0.301; HR = 0.50, 95% C.I. [0.14–1.79]; *p* = 0.288) (Figure 2a). Finally, among patients with stage IV (303 patients, 76 exposed to antiplatelet therapy), a total of 82 cases of brain metastases were diagnosed, of which only 9 belonged to the antiplatelet-exposed group (chi-square; *p* = 0.001; HR = 0.380, 95% C.I. [0.19–0.76]; *p* = 0.006) (Figure 2a).

Similarly, the mean time to brain metastasis onset varied between exposure groups depending on tumor stage. Among patients not receiving antiplatelet therapy, this time differed significantly across tumor stages (Figure 2b), whereas in antiplatelet-treated patients, these stage-based differences were less pronounced (Figure 2c). Focusing on patients with stage IV, the mean time of occurrence of brain metastases in the group of patients with antiplatelet therapy was significantly longer than in patients without this medication (62.4 vs. 34.5 months; log-rank, *p* = 0.002).

### 3.2. Antiplatelet Therapy Reduces the Risk of Presenting Brain Metastases at Diagnosis

If we consider only patients with brain metastases at the time of NSCLC diagnosis (62 cases), only eight of them (12.9%) had been exposed to antiplatelet therapy (chi-square; *p* = 0.014) for a mean exposure time of 49.25 months (SD = 40.83). This duration of antiplatelet use showed no statistically significant difference when compared to patients without brain metastases at diagnosis (51.4 months; SD = 36.7; Mann–Whitney U, *p* = 0.803).

Additionally, binary logistic regression analysis was performed to analyze possible risk factors associated with the presence of brain metastases at the time of NSCLC diagnosis. Univariate analysis showed that both age (OR = 0.968; 95% C.I. [0.946–0.991]; *p* = 0.006) and exposure to antiplatelets before diagnosis (OR = 0.396; 95% C.I. [0.185–0.851]; *p* = 0.018) were protective factors for the presence of brain metastases at the time of diagnosis (Table 3). However, in the multivariate analysis, although there was a clear trend toward significance, antiplatelet use did not reach statistical significance (*p* = 0.068; Table 3).

### 3.3. Antiplatelet Therapy Reduces the Risk of Developing Brain Metastases Throughout the Disease, Especially in Patients with Advanced Stages at Diagnosis

After excluding patients with metastases at the time of diagnosis (62 patients) and those who did not receive active treatment after NSCLC diagnosis (141 patients), a comparative analysis was conducted between patients with and without exposure to antiplatelet therapy. Like the previous comparison of these groups including all patients, those exposed to antiplatelet therapy had a higher prevalence of cardiovascular (hypertension, heart failure, and renal) and metabolic comorbidities (diabetes and dyslipidemia) (chi-square, *p* < 0.05) (Table 4). Additionally, the mean age of this patient group was higher (70.27 vs. 64.44; Mann–Whitney U, *p* < 0.001). Finally, the distribution of stages was significantly different between the two patient groups (Table 4). Specifically, the main differences between patients without and with antiplatelet use were in the percentage of patients with stage 0 and I (25.5% vs. 43.4%, respectively) and with stage IV (38.4% vs. 25.9%, respectively) (chi-square, *p* = 0.001). Consequently, there were significant differences in the types of treatments received (chi-square, *p* = 0.020).

Regarding the development of brain metastases, 33 cases (10.9%) in the group of patients using antiplatelets developed this complication, compared to 6 cases (4.2%) in the group of patients not using antiplatelets. This difference reached statistical significance (chi-square, *p* = 0.019).

As in the previous analyses, a stage-by-stage assessment was performed to explore the association between antiplatelet use and the development of brain metastases. Among patients with stages 0 and I, two patients developed brain metastases (one in each group [1.3% vs. 1.6%]) (chi-square, *p* = 0.868; HR = 1.28, 95% C.I. [0.08–20.59]; *p* = 0.858) (Figure 3a). In stage II, four patients (12.1%) developed brain metastases (three in the non-antiplatelet group [12.0% vs. 12.5%]) (chi-square, *p* = 0.970; HR = 1.07, 95% C.I. [0.11–10.31]; *p* = 0.954) (Figure 3a). In stage III, 15 patients developed brain metastases, with 12 cases in the non-antiplatelet group [14.3% vs. 8.3%] (chi-square, *p* = 0.366; HR = 0.57, 95% C.I. [0.16–2.03]; *p* = 0.387) (Figure 3a). Finally, in stage IV, 18 patients developed brain metastases, with 17 of them in the non-antiplatelet group [14.7% vs. 2.7%] (chi-square, *p* = 0.049; HR = 0.236, 95% C.I. [0.03–1.77]; *p* = 0.161) (Figure 3a).

The mean time to the onset of brain metastases differed in the antiplatelet exposure groups, depending on the tumor stage. For patients not exposed to antiplatelet therapy, the onset period of brain metastases showed significant differences between different tumor stages (log-rank, *p* < 0.001) (Figure 3b), while such differences were not identified in the group of patients exposed to antiplatelet therapy (log-rank, *p* = 0.328) (Figure 3c). The mean time for the development of brain metastases in patients with stage IV not exposed to antiplatelet therapy was 46.0 months, compared to 59.4 months in the group of patients exposed to antiplatelet therapy (log-rank test, *p* = 0.127).

### 3.4. Initiating Antiplatelet Therapy After the Diagnosis of NSCLC May Have a Positive Effect in Preventing the Development of Brain Metastases

A comparative analysis was conducted between patients who were never exposed to antiplatelet therapy (*n* = 302) and those who initiated exposure after the diagnosis of NSCLC (*n* = 34; mean time to initiation after diagnosis of 2.98 months [SD = 12.2]). Patients with brain metastases at diagnosis, those who did not receive treatment after diagnosis, and those exposed to antiplatelet therapy for at least more than a month before the diagnosis of the disease were excluded from this analysis. In this patient group, a total of 34 cases with brain metastases were diagnosed. All cases belonged to the group not using antiplatelet therapy (chi-square, *p* = 0.032). Subgroup analysis by tumor stage was not carried out due to the significant difference in group sizes for comparison.

## 4. Discussion

This retrospective observational study suggests an association between antiplatelet therapy and a reduced incidence of brain metastases in NSCLC. Patients receiving antiplatelet agents exhibited a lower risk of developing brain metastases throughout the disease course and a longer time to their occurrence compared to those not exposed to this therapy. This effect was particularly pronounced in patients with advanced disease, especially in stage IV. Additionally, patients exposed to antiplatelet therapy before NSCLC diagnosis had a lower risk of presenting brain metastases at diagnosis. Furthermore, in patients who initiated antiplatelet therapy after NSCLC diagnosis, no cases of brain metastases were recorded, suggesting a potential additional benefit in preventing metastatic spread. These findings support the hypothesis of a protective role of antiplatelet therapy in tumor progression in NSCLC and highlight the need for further research to better understand its impact across different patient subgroups.

A growing body of evidence suggests that antiplatelet therapy may play a role in modulating cancer progression and metastatic dissemination, although its impact remains incompletely understood. A systematic review comparing observational studies and randomized trials found that regular aspirin use was associated with a reduced risk of metastasis in several cancer types, including colorectal, esophageal, and breast cancer [45]. Notably, the proportion of cancers with distant metastases was lower in aspirin users, reinforcing the hypothesis that platelet inhibition may interfere with tumor dissemination. These findings are consistent with our observations in NSCLC, where prior exposure to antiplatelet therapy was associated with a lower incidence of brain metastases and a prolonged time to their development. On the other hand, a large-scale, population-based cohort study in Taiwan investigated the association between long-term, low-dose aspirin use and the risk of primary cancer in survivors of ischemic cardiac or cerebrovascular disease (ICCD) [46]. The study demonstrated a significant reduction in cancer incidence, particularly with prolonged aspirin use, with protective effects observed across multiple tumor types, including lung cancer. While these findings support the potential role of platelet inhibition in mitigating tumor progression, they do not directly address whether antiplatelet therapy influences metastatic spread, particularly to the brain in NSCLC patients. Given the high burden of brain metastases in NSCLC and the limited data evaluating the impact of antiplatelet agents in this specific setting, further investigation was necessary to clarify whether these therapies could have a protective effect against metastatic dissemination.

Our study’s findings align with the existing literature suggesting that antiplatelet therapy may play a role in reducing cancer metastasis. Experimental data indicate that lowering platelet count can reduce tumor growth and metastasis [40]. Based on the mechanisms by which platelets contribute to cancer progression, it is conceivable that drugs reducing platelet count or platelet activation might attenuate cancer progression and improve outcomes [40]. As discussed in the introduction, the effects of antiplatelet agents on tumor progression are multifaceted and may involve mechanisms at different biological levels. On the one hand, these drugs could directly act on the tumor cell, inhibiting the activity of certain molecules that would otherwise promote a more aggressive phenotype [48,49]. On the other hand, the antitumor effect of antiplatelet agents could be linked to their activity on platelets themselves. The ability of tumor cells to activate platelets and induce the formation of platelet aggregates associated with tumor cells (tumor cell-induced platelet aggregation [TCIPA]) is well-known [50]. TCIPA provides protection to circulating tumor cells, both against intravascular mechanical forces and the immune system, creating a physical barrier around the tumor cell [32]. In our study, patients exposed to antiplatelet therapy showed a significantly lower risk of developing brain metastases throughout the disease (6.9% vs. 20.0%, *p* < 0.001), supporting the hypothesis that antiplatelet therapy disrupts platelet–tumor interactions critical for metastatic dissemination. This protective effect was particularly evident in advanced stages, where the risk of brain metastases was markedly reduced (stage IV: HR = 0.38, *p* = 0.006).

These findings suggest that antiplatelet therapy could modulate the tumor microenvironment, impacting processes such as angiogenesis and the formation of premetastatic niches [51]. Platelets release proangiogenic factors, such as vascular endothelial growth factor (VEGF) and angiopoietin-1, which are critical for neovascularization and tumor progression. By inhibiting platelet activation, antiplatelet agents could reduce the availability of these factors, potentially impairing the establishment of metastatic sites. In our cohort, patients using antiplatelet therapy not only had a lower incidence of brain metastases at diagnosis (3.9% vs. 12.1%, *p* = 0.014) but also experienced a longer mean time to the development of metastases during follow-up (77.5 vs. 62.6 months, *p* < 0.001), suggesting a potential role in delaying the formation of premetastatic niches in the brain.

Beyond their influence on platelet activation and angiogenesis, antiplatelet agents may also play a role in modulating immune responses to tumor cells. Platelets have been shown to shield circulating tumor cells (CTCs) from immune recognition by natural killer (NK) cells through the expression of major histocompatibility complex class I (MHC-I)-like molecules [52]. Although our study did not evaluate immune modulation directly, this mechanism warrants further exploration in NSCLC, particularly given the central role of immunotherapy in the treatment of this disease. Future studies should investigate whether antiplatelet therapy could enhance the efficacy of immunotherapy by reducing platelet-mediated immune evasion and facilitating a stronger antitumor immune response.

Finally, antiplatelet therapy may interact synergistically with conventional cancer treatments such as chemotherapy and immunotherapy. Platelet-derived factors have been implicated in creating a supportive microenvironment for tumor cells, often reducing the effectiveness of systemic treatments [53]. The longer time to metastasis development observed in our study may partially reflect this synergy, as antiplatelet therapy might enhance the efficacy of systemic treatments, particularly in patients with advanced disease who are more likely to receive multimodal therapies.

Overall, the results of this study underscore the potential of antiplatelet agents as a complementary strategy to mitigate brain metastasis risk in NSCLC. By targeting TCIPA, modulating the tumor microenvironment, and synergizing with existing treatments, antiplatelet agents could improve metastatic outcomes, particularly in patients with advanced-stage disease where brain metastases are a common and devastating complication.

However, it is important to note that while preclinical and observational studies provide supportive evidence, randomized controlled trials are necessary to establish a definitive causal relationship between antiplatelet therapy and reduced metastasis in NSCLC. Additionally, the potential risks associated with antiplatelet therapy, such as bleeding complications, must be carefully weighed against the potential benefits. In the context of NSCLC, brain metastases are a common and serious complication. A study focusing on patients with metastatic brain tumors, including a significant proportion with NSCLC, found that the use of antiplatelet agents was not associated with an increased risk of intracranial hemorrhage [54].

This study has some limitations that should be acknowledged. First, its retrospective design inherently carries certain biases, such as variability in clinical documentation and the absence of randomization. However, the consistency of our findings across different analyses, particularly in advanced-stage patients, strengthens the validity of the observed association between antiplatelet therapy and reduced brain metastasis risk. Additionally, while the sample size of specific subgroups, such as those with molecular alterations, was relatively small, the overall cohort was sufficiently large to detect significant differences in brain metastasis incidence and progression-free survival. The heterogeneity in the determination of molecular markers (e.g., EGFR, ALK, and ROS1) further complicates identifying subgroups of patients who might benefit the most from antiplatelet therapy, warranting further studies with standardized molecular profiling. While there were differences in tumor stage distribution between groups, with a higher proportion of early-stage patients in the antiplatelet cohort, this does not fully explain the observed protective effect, as the reduction in brain metastases remained significant even when analyzing advanced-stage patients separately. Another potential limitation is that the study was conducted in a single center, which could affect the generalizability of the findings. Nevertheless, the clinical characteristics of our cohort align with those reported in broader NSCLC populations, supporting the relevance of our results. Finally, while various antiplatelet agents were used, the small sample size and heterogeneity precluded an analysis of the relative efficacy of different agents, leaving this question unanswered for future research.

## 5. Conclusions

In conclusion, this study suggests that antiplatelet therapy is associated with a reduced risk of brain metastases in NSCLC, both at the time of diagnosis and during disease progression, with the effect being particularly notable in advanced stages. These findings suggest that antiplatelet agents could play a complementary role in mitigating metastatic risk and improving outcomes in this population. Clinically, this underscores the potential for integrating antiplatelet therapy into multimodal treatment strategies for NSCLC, especially in combination with standard therapies such as immunotherapy, which is now a cornerstone in NSCLC management. Future research should aim to confirm these results through prospective randomized trials and explore the biological mechanisms underlying these effects, including their role in modulating the tumor microenvironment, immune response, and interactions with other therapies. Additionally, studies should identify the optimal type, timing, and duration of antiplatelet therapy to maximize its benefit while minimizing risks.

## Figures and Tables

**Figure 1 cancers-17-02059-f001:**
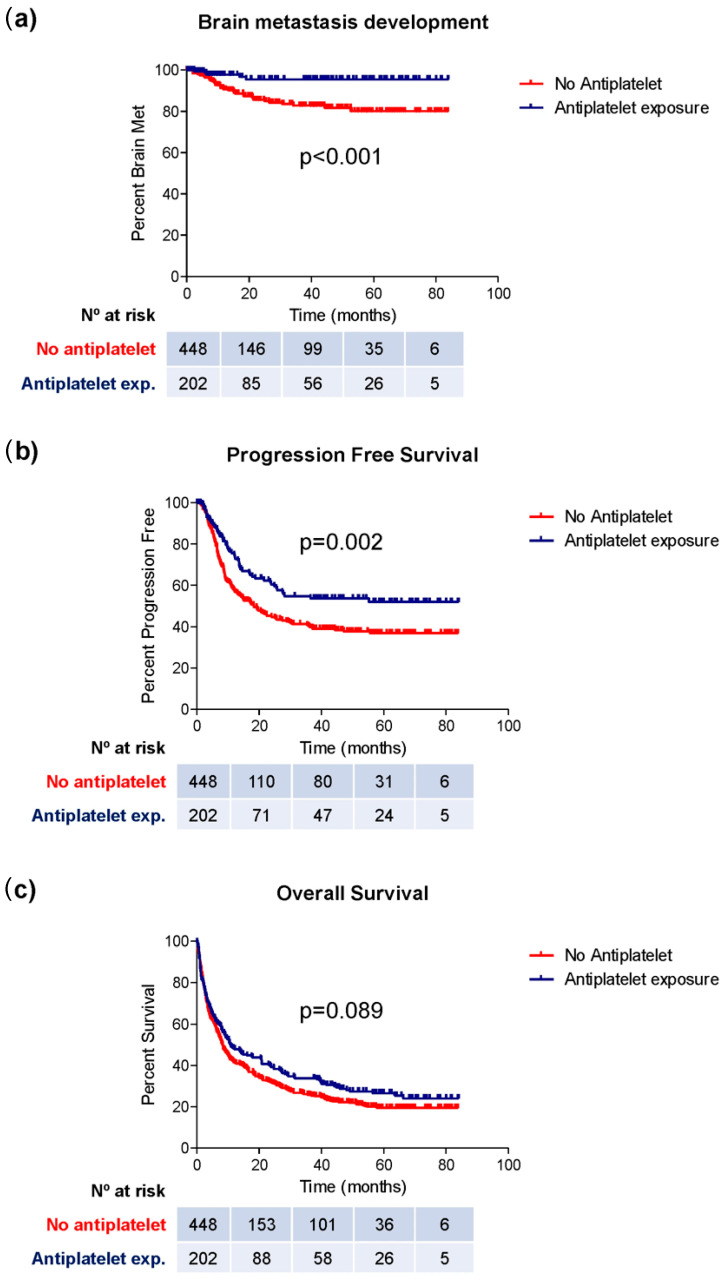
Kaplan–Meier curves representing the median period of brain metastasis development (**a**), PFS (**b**), and OS (**c**) in groups of patients exposed or not exposed to antiplatelet therapy.

**Figure 2 cancers-17-02059-f002:**
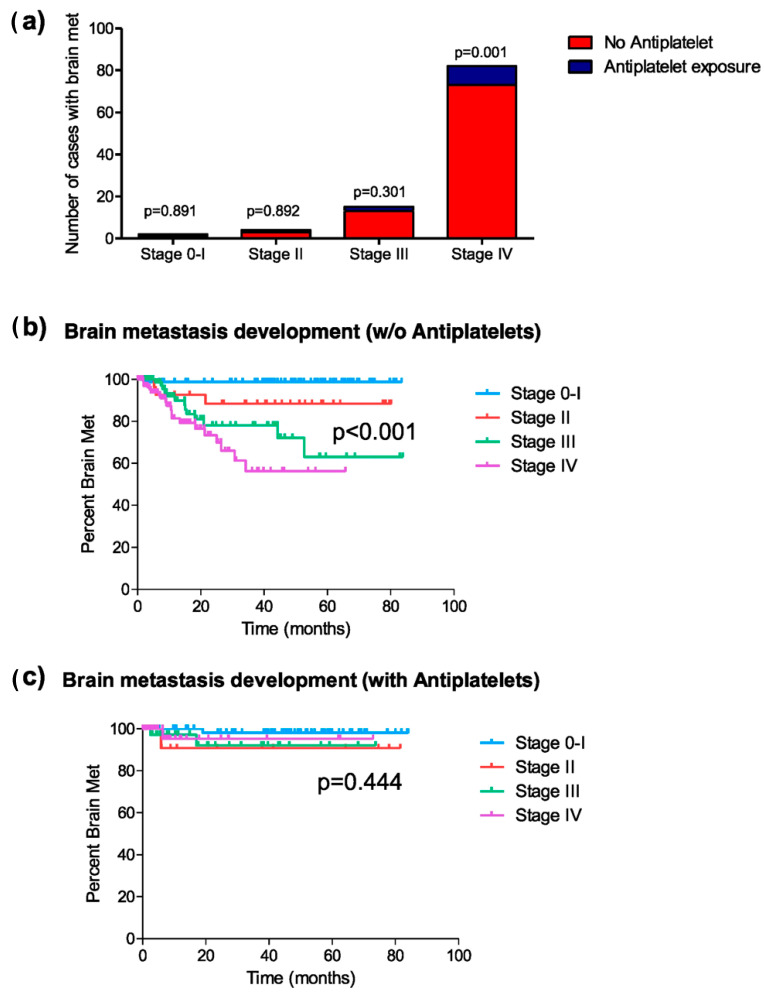
Brain metastasis development at any time during the disease. (**a**) Distribution of cases of brain metastasis in patients exposed or not exposed to antiplatelet therapy, classified by tumor stage; (**b**) Kaplan–Meier curves representing the median period of brain metastasis development at each tumor stage in patients not exposed to antiplatelet therapy; and (**c**) those exposed to antiplatelet therapy during the disease.

**Figure 3 cancers-17-02059-f003:**
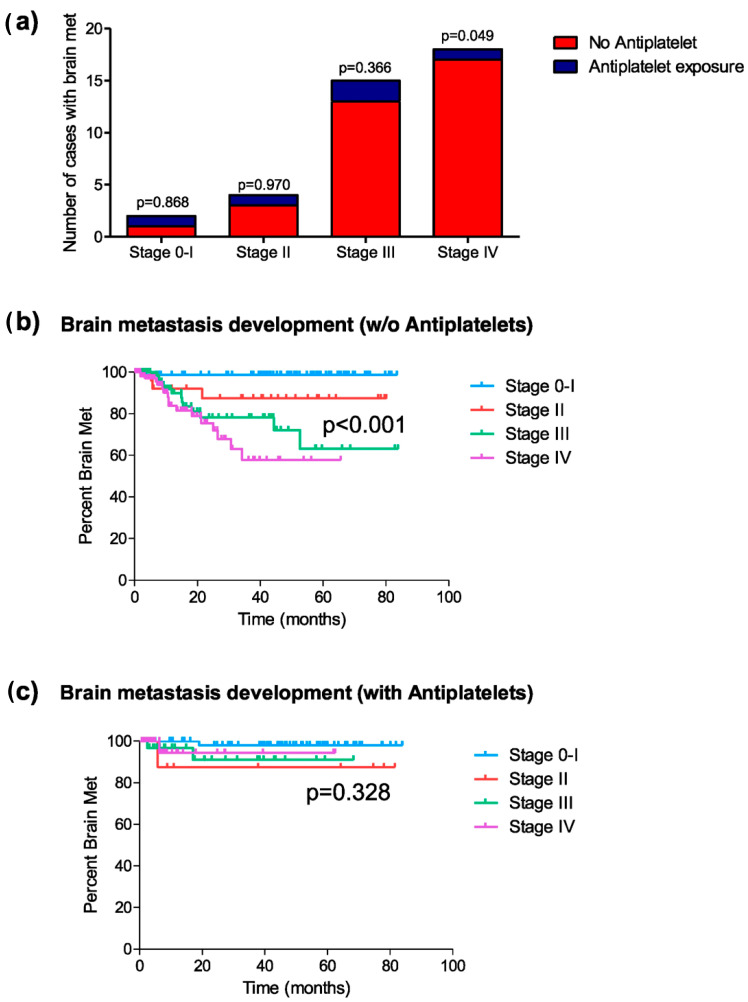
Development of brain metastasis post-diagnosis (excluding patients who did not receive treatment). (**a**) Distribution of cases of brain metastasis in patients exposed or not to platelet anti-aggregation, classified by tumor stage (chi-square); (**b**) Kaplan–Meier curves representing the mean period of brain metastasis development in each tumor stage for patients not exposed to platelet anti-aggregation; and (**c**) those exposed to platelet anti-aggregation during the disease.

**Table 1 cancers-17-02059-t001:** Initial characteristics of patients included in the study.

Variable		Mean (SD) Count (%)
Age at diagnosis		67.23 (SD = 10.94)
Gender (Men:Women)		459:192
Hypertension		313 (48.1%)
Diabetes		169 (26.0%)
Cardiac failure		87 (13.4%)
Renal failure		37 (5.7%)
Hypercholesterolemia		252 (38.7%
COPD		189 (29.0%)
Pack-Year		45.81 (SD = 31.55)
Initial symptom	Asymptomatic	264 (40.6%)
	Respiratory	212 (32.6%)
	Respiratory and worsening in general condition	50 (7.7%)
	Neurological	40 (6.1%)
	Pain	40 (6.1%)
	Worsening in general condition	36 (5.5%)
	Superior large vein syndrome	1 (0.2%)
	Aphonia	2 (0.3%)
	Dysphagia	3 (0.5%)
	Pleural effusion	3 (0.5%)
Histological diagnosis	Adenocarcinoma	413 (63.4%)
	Epidermoid	178 (27.3%)
	Other	60 (9.2%)
EGFR mutation (*n* = 398)		55 (12.4%)
ALK mutation (*n* = 364)		9 (2.2%)
ROS1 expression (*n* = 168)		1 (0.6%)
PDL1 expression (*n* = 121)		52 (39.4%)
BRAF mutation (*n* = 51)		5 (9.8%)
Pathological staging	Stage 0 and I	148 (22.8%)
	Stage II	43 (6.6%)
	Stage III	154 (23.8%)
	Stage IV	303 (46.8%)
Treatment	Surgery	114 (17.5%)
	Chemotherapy (CT)	203 (31.2%)
	Radiotherapy (RT)	26 (4.0%)
	CT + RT	60 (9.2%)
	Palliative	119 (18.3%)
	Surgery + CT	61 (9.4%)
	Surgery + CT + RT	6 (0.9%)
	Surgery + RT	2 (0.3%)
	No treatment	40 (6.2%)
	Laser therapy	1 (0.2%)
	Follow-up	4 (0.6%)
Brain metastasis development		103 (15.8%)
Brain metastasis at diagnosis		62 (9.5%)

**Table 2 cancers-17-02059-t002:** Comparison between patients with or without antiplatelet use.

		Non-Use of Antiplatelets (*n* = 446)	Use of Antiplatelets (*n* = 204)	*p*-Value
Variable		Mean (SD) Count (%)	Mean (SD) Count (%)	
Age at diagnosis		65.19 (SD = 10.97)	71.66 (SD = 9.51)	<0.001
Gender (Men:Women)		300:146	158:46	0.009
Hypertension		175 (39.2%)	137 (67.2%)	<0.001
Diabetes		89 (20.0%)	79 (38.7%)	<0.001
Cardiac failure		29 (6.5%)	57 (27.9%)	<0.001
Renal failure		15 (3.4%)	22 (10.8%)	<0.001
Hypercholesterolemia		144 (32.3%)	107 (52.5%	<0.001
COPD		122 (27.4%)	67 (32.8%)	0.163
Pack-Year		44.93 (SD = 31.31)	47.89 (SD = 32.08)	0.220
Initial symptom	Asymptomatic	175 (39.2%)	88 (43.1%)	0.442
	Respiratory	148 (33.2%)	64 (31.4%)	
	Respiratory and worsening in general condition	35 (7.8%)	15 (7.4%)	
	Neurological	33 (7.4%)	7 (3.4%)	
	Pain	28 (6.3%)	12 (5.9%)	
	Worsening in general condition	23 (5.2%)	13 (6.4%)	
	Superior large vein syndrome	1 (0.2%)	-	
	Aphonia	1 (0.2%)	1 (0.5%)	
	Dysphagia	1 (0.2%)	2 (1.0%)	
	Pleural effusion	1 (0.2%)	2 (1.0%)	
Histological diagnosis	Adenocarcinoma	285 (63.9%)	127 (62.3%)	0.809
	Epidermoid	122 (27.4%)	56 (27.5%)	
	Other	39 (8.7%)	21 (10.3%)	
EGFR mutation (*n* = 398)		40 (13.1%)	15 (10.9%)	0.539
ALK mutation (*n* = 364)		7 (2.5%)	2 (1.6%)	0.726
ROS1 expression (*n* = 168)		1 (0.8%)	-	1.000
PDL1 expression (*n* = 121)		34 (36.2%)	17 (45.9%)	0.325
BRAF mutation (*n* = 51)		4 (9.8%)	2 (13.3%)	0.654
Pathological staging	Stage 0 and I	80 (18.1%)	67 (32.8%)	<0.001
	Stage II	31 (7.0%)	12 (5.9%)	
	Stage III	105 (23.7%)	49 (24.0%)	
	Stage IV	227 (51.2%)	76 (37.3%)	
Treatment	Surgery	72 (16.2%)	41 (20.1%)	0.026
	Chemotherapy (CT)	154 (34.6%)	49 (24.0%)	
	Radiotherapy (RT)	11 (2.5%)	15 (7.4%)	
	CT + RT	52 (11.7%)	21 (10.3%)	
	Palliative	76 (17.1%)	43 (21.1%)	
	Surgery + CT	40 (9.0%)	20 (9.8%)	
	Surgery + CT + RT	3 (0.7%)	3 (1.5%)	
	Surgery + RT	2 (0.4%)	-	
	No treatment	31 (7.0%)	9 (4.4%)	
	Laser therapy	-	1 (0.5%)	
	Follow-up	2 (0.4%)	2 (1.0%)	
Brain metastasis development		89 (20.0%)	14 (6.9%)	<0.001
Brain metastasis at diagnosis		54 (12.1%)	8 (3.9%)	<0.001
Brain metastasis period (mean, months)		62.6 [58.7–66.6]	77.5 [74.0–81.0]	<0.001
Progression-free survival (mean, months)		38.7 [34.3–43.2]	49.9 [43.5–56.3]	0.002
Overall Survival (median, months)		8.2 [6.8–9.6]	10.6 [5.8–15.3]	0.089

**Table 3 cancers-17-02059-t003:** Binary logistic regression analysis to examine risk factors associated with the presence of brain metastasis at the diagnosis of the disease.

Univariate							
Variable		B	S.E.	OR	95% C.I. for EXP(B)		Sig.
					Lower	Upper	
Age		−0.033	0.012	0.968	0.946	0.991	0.006
Gender (male)		−0.149	0.286	0.602	0.491	1.510	0.602
Pack-Year		0.004	0.004	1.004	0.996	1.004	0.345
COPD		−0.275	0.310	0.759	0.414	1.394	0.375
Hypertension		−0.426	0.273	0.653	0.382	1.117	0.120
Diabetes		−0.205	0.318	0.815	0.437	1.519	0.519
Cardiac failure		−0.868	0.530	0.420	0.148	1.187	0.102
Renal failure		−0.641	0.740	0.527	0.124	2.244	0.527
Hypercholesterolemia		−0.227	0.281	0.797	0.459	1.384	0.421
Histology	Adenocarcinoma	0.466	0.297	1.594	0.890	2.854	0.117
	Epidermoid	−0.494	0.334	0.610	0.317	1.175	0.139
	Other	−0.162	0.487	0.850	0.327	2.210	0.739
EGFR mutation		0.210	0.437	1.234	0.524	2.907	0.630
ALK mutation		-	-	-	-	-	-
ROS1 mutation		-	-	-	-	-	-
PDL1 expression		−0.288	0.640	0.750	0.214	2.630	0.653
BRAF mutation		0.588	1.193	1.800	0.174	18.638	0.622
Antiplatelet use		−0.926	0.390	0.396	0.185	0.851	0.018
Multivariate							
Variable		B	S.E.	OR	95% C.I. for EXP(B)		Sig.
					Lower	Upper	
Age		−0.026	0.012	0.974	0.951	0.998	0.033
Antiplatelet use		−0.732	0.401	0.481	0.219	1.056	0.068

**Table 4 cancers-17-02059-t004:** Comparative analysis between patients exposed or not to platelet anti-aggregation, having excluded those patients who had brain metastasis at the time of diagnosis and those who did not receive active treatment.

		Non-Use of Antiplatelets (*n* = 302)	Use of Antiplatelets (*n* = 143)	*p*-Value
Variable		Mean (SD) Count (%)	Mean (SD) Count (%)	
Age at diagnosis		64.44 (SD = 10.99)	70.27 (SD = 9.03)	<0.001
Gender (Men:Women)		203:99	108:35	0.078
Hypertension		120 (39.7%)	96 (67.1%)	<0.001
Diabetes		60 (19.9%)	58 (40.6%)	<0.001
Cardiac failure		20 (6.6%)	36 (25.2%)	<0.001
Renal failure		11 (3.6%)	15 (10.5%)	0.008
Hypercholesterolemia		102 (33.8%)	80 (55.9%	<0.001
COPD		81 (26.8%)	47 (32.9%)	0.217
Pack-Year		43.25 (SD = 32.26)	49.14 (SD = 31.44)	0.038
Initial symptom	Asymptomatic	143 (47.4%)	76 (53.1%)	0.548
	Respiratory	102 (33.8%)	41 (28.7%)	
	Respiratory and worsening in general condition	21 (7.0%)	12 (8.4%)	
	Neurological	4 (1.3%)	1 (0.7%)	
	Pain	17 (5.6%)	7 (4.9%)	
	Worsening in general condition	12 (4.0%)	3 (2.1%)	
	Superior large vein syndrome	1 (0.3%)	-	
	Aphonia	1 (03%)	-	
	Dysphagia	-	1 (0.7%)	
	Pleural effusion	1 (0.3%)	2 (1.4%)	
Histological diagnosis	Adenocarcinoma	195 (64.6%)	85 (59.2%)	0.501
	Epidermoid	81 (26.8%)	46 (32.2%)	
	Other	26 (8.6%)	12 (8.4%)	
EGFR mutation (*n* = 307)		33 (15.6%)	11 (11.5%)	0.383
ALK mutation (*n* = 281)		6 (3.1%)	2 (2.3%)	1.000
ROS1 expression (*n* = 132)		1 (1.0%)	-	1.000
PDL1 expression (*n* = 101)		28 (39.4%)	14 (46.7%)	0.516
BRAF mutation (*n* = 38)		2 (7.7%)	1(8.3%)	1.000
Pathological staging	Stage 0 and I	77 (25.5%)	62 (43.4%)	0.001
	Stage II	25 (8.3%)	8 (5.6%)	
	Stage III	84 (27.8%)	36 (25.2%)	
	Stage IV	116 (38.4%)	37 (25.9%)	
Treatment	Surgery	72 (16.2%)	41 (28.7%)	0.020
	Chemotherapy (CT)	128 (42.4%)	44 (30.8%)	
	Radiotherapy (RT)	11 (3.6%)	15 (10.5%)	
	CT + RT	49 (16.2%)	21 (14.7%)	
	Surgery + CT	39 (12.9%)	19 (13.3%)	
	Surgery + CT + RT	2 (0.7%)	3 (2.1%)	
	Surgery + RT	2 (0.7%)	-	
Brain metastasis development		33 (10.9%)	6 (4.2%)	0.019
Progression-free survival (mean, months)		40.0 [35.4–44.7]	50.2 [43.4–57.0]	0.012
Overall Survival (median, months)		18.2 [13.7–22.7]	26.5 [18.3–34.7]	0.216

## Data Availability

The data supporting the findings of this study are available from the corresponding author upon reasonable request.

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
