# Peer review of "Antiplatelet Therapy Mitigates Brain Metastasis Risk in Non-Small Cell Lung Cancer: Insights from a Comprehensive Retrospective Study"

_cancers, 2025, doi:10.3390/cancers17132059_

Round 1

Reviewer 1 Report

Comments and Suggestions for Authors

Overall evaluation. Given their pro-metastatic role, targeting platelet function has emerged as a potential therapeutic strategy to limit metastasis and improve clinical outcomes. It has been acknowledged that aspirin use is associated with a reduced risk of metastasis in several cancer types, leading to the hypothesis that platelet inhibition may interfere with tumor dissemination. The impact of antiplatelet therapy in real-world NSCLC populations—including its potential association with metastatic burden at diagnosis—is systematically explored in this retrospective, single-center observational study. The primary objective was to evaluate the percentage of NSCLC patients who developed brain metastases during the course of the disease. The authors observed that the development of brain metastases was significantly higher in the group not using antiplatelets (20%) compared to 6.9% in the antiplatelet-exposed group. Subgroup analyses by tumor stage revealed that, in patients with stage IV NSCLC, antiplatelet therapy was associated with a significantly reduced incidence of brain metastases (p = 0.001) and a longer mean time to their development (p = 0.002). The authors have conducted a well-designed and methodologically sound real-world analysis to address their hypothesis. Given the real-world data study design, multivariate analyses are essential to account for potential confounders. The authors performed binary univariate and multivariate regression analyses. However, for Chapter 3.4, an additional multivariate analysis would be necessary to confirm the reported findings. The conclusion would benefit from a more cautious phrasing to ensure that the strength of the evidence is not overstated.

Comment 1: Methods: Please provide a more detailed explanation of the binary univariate and multivariate regression analyses performed. Specifically, clarify whether the binary outcome variable was: presence vs. absence of brain metastases. Additionally, please specify which covariates were included in the multivariate model.

Comment 2: Results (Page 15, Chapter 3.4): You mention that a subgroup analysis was not conducted due to differences in group sizes. However, to my knowledge, it is still feasible to perform a binary multivariate regression analysis, similar to the approach used in Table 3. I encourage you to conduct such an analysis incorporating potential confounders, such as age, tumor stage, and cardiovascular and metabolic comorbidities. This would help determining whether the observed effect of antiplatelet therapy on brain metastasis remains statistically significant after adjustment for these variables.

Comment 3. Conclusion: Please replace the term “demonstrates” with a more cautious alternative such as “suggests.” As the association between antiplatelet therapy and reduced brain metastases has only been confirmed in your univariate and not multivariate analyses (as shown in Table 3), the current wording may overstate the strength of the evidence.

Author Response

1. Methods: Please provide a more detailed explanation of the binary univariate and multivariate regression analyses performed. Specifically, clarify whether the binary outcome variable was: presence vs. absence of brain metastases. Additionally, please specify which covariates were included in the multivariate model.

Thank you very much for this comment. We have added further details in the Methods section to clarify that the outcome variable was the presence vs. absence of brain metastases. We have also specified that no additional covariates were included beyond those that were statistically significant in the univariate analysis.

2. Results (Page 15, Chapter 3.4): You mention that a subgroup analysis was not conducted due to differences in group sizes. However, to my knowledge, it is still feasible to perform a binary multivariate regression analysis, similar to the approach used in Table 3. I encourage you to conduct such an analysis incorporating potential confounders, such as age, tumor stage, and cardiovascular and metabolic comorbidities. This would help determining whether the observed effect of antiplatelet therapy on brain metastasis remains statistically significant after adjustment for these variables.

Thank you for your suggestion to perform a binary multivariate regression analysis adjusting for potential confounders such as age, tumor stage, and comorbidities. However, we would like to clarify that this analysis is not feasible with our dataset. In the analysis focused exclusively on brain metastases diagnosed during follow-up (as cases present at diagnosis were excluded), all events occurred exclusively in the group of patients who did not receive antiplatelet therapy. This means that there were no events in the group exposed to antiplatelet therapy (a situation known as "complete separation"), which prevents the convergence and parameter estimation in logistic regression models.

For this reason, we believe that the most appropriate approach in this scenario was the chi-square test, which already demonstrated a statistically significant association. Unfortunately, a multivariate model would not provide additional or reliable information given these limitations.

We appreciate your comment and remain at your disposal for any further clarification.

3. Conclusion: Please replace the term “demonstrates” with a more cautious alternative such as “suggests.” As the association between antiplatelet therapy and reduced brain metastases has only been confirmed in your univariate and not multivariate analyses (as shown in Table 3), the current wording may overstate the strength of the evidence.

Thank you for this observation. We have taken your suggestion into account.

Reviewer 2 Report

Comments and Suggestions for Authors

This is a very relevant retrospective study in patients with NSCLC that aimed to assess if antiplatelet exposure was associated with a lower risk of BM. Authors demostrate that in the bivariate analysis the use of antiplatelets before the diagnosis of BM was associated with a lower risk of BM and a longer brain metastases free survival. Unfortunately, in the multivariate analysis this was not statistically significant; but probably it dos have clinical relavance. A prospective analysis would be ideal to answer this question. Nevertheles, this is a valuable study.

Minor areas that could aid the manuscript are:

Even if in the multivariable tests the variable is not "significant" it does not mean it lacks clinical evidence, collinearity and correlation interactions should be included in the discussion. Authors do have the vraible "time" did they consider using Cox-regression analysis instead of logistic regression to adjustment?

There are many manuscripts describing variables associated with the risk of BM, please consider them (i.e., doi: 10.1007/s12032-014-0228-9., doi: 10.1007/s11060-021-03849-w, )

Number at risk vary in Figure 1, please explain

Correcting minor typo errors

A proofreading and English editing could improve soundness and understanding, mainly for the manuscript uses past, present and future tenses in the same paragraphs.

Comments on the Quality of English Language

Very valuable manuscript, minor corrections are presented

Author Response

1. Even if in the multivariable tests the variable is not "significant" it does not mean it lacks clinical evidence, collinearity and correlation interactions should be included in the discussion. Authors do have the variable "time" did they consider using Cox-regression analysis instead of logistic regression to adjustment?

Thank you for this suggestion. However, as detailed in the manuscript, the binary logistic regression was specifically conducted to assess the risk of presenting with brain metastases at the time of diagnosis. Consequently, the “time” variable is not relevant for this particular analysis.

On the other hand, the Cox regression model was employed in the subgroup analysis by tumor stage to evaluate the effect of antiplatelet therapy on the development of brain metastases over the course of the disease.

2. There are many manuscripts describing variables associated with the risk of BM, please consider them (i.e., doi: 10.1007/s12032-014-0228-9., doi: 10.1007/s11060-021-03849-w, )

Thank you for this suggestion. The references have been added to the Introduction section.

3. Number at risk vary in Figure 1, please explain.

This was an oversight, which has now been corrected.

4. A proofreading and English editing could improve soundness and understanding, mainly for the manuscript uses past, present and future tenses in the same paragraphs.

Thank you for this observation. We have carefully revised the manuscript, with particular attention to the consistency of verb tenses. In addition, the text has been reviewed by a native English speaker to improve clarity and fluency.

Reviewer 3 Report

Comments and Suggestions for Authors

Authors concluded that taking antiplatelet medicine might help lower the chance of brain metastases in patients with non-small cell lung cancer, both when the cancer is first found and as it gets worse. This effect seems to be especially strong in more advanced stages of the disease. Although this is a retrospective study from a single center, I think it is a well-thought-out paper.

I have one question. When and how were the brain metastases discovered? Did authors check MRI after symptoms started, or periodically? In the methods section, I think authors should provide more details about the staging procedure of initial diagnosis and diagnosis of brain metastases after disease progression.

Figure 2b and 3b, wo Antiplatelets -> w/o Antiplatelets?

Author Response

1. I have one question. When and how were the brain metastases discovered? Did authors check MRI after symptoms started, or periodically? In the methods section, I think authors should provide more details about the staging procedure of initial diagnosis and diagnosis of brain metastases after disease progression.

Thank you for your question. We agree that additional details were needed to clarify how brain metastases were diagnosed. As described in the revised Methods section, brain magnetic resonance imaging (MRI) was performed when patients presented with neurological symptoms suggestive of central nervous system involvement. Once brain metastases were detected, follow-up brain MRIs were conducted periodically. In cases where brain MRI was included in the initial staging work-up, follow-up MRIs were also performed at regular intervals, regardless of symptoms.

2. Figure 2b and 3b, wo Antiplatelets -> w/o Antiplatelets?

This error has been corrected.